# Transitions in sexual behaviour among gay, bisexual, and other men who have sex with men in England: Data from a prospective study

**Nadia Hanum**[1], **Valentina Cambiano**[1], **Janey Sewell**[1], **Alison J. Rodger**[1], **David Asboe**[2], **Gary Whitlock**[2], **Richard Gilson**[1,3], **Amanda Clarke**[4], **Ada R. Miltz**[1], **Simon Collins**[5], **Andrew N. Phillips**[1], **Fiona C. Lampe**[1]*, for the AURAH2 Study Group¶

1 UCL Institute for Global Health, London, United Kingdom, 2 Chelsea and Westminster Hospital NHS Foundation Trust, London, United Kingdom, 3 Central and North West London NHS Foundation Trust, London, United Kingdom, 4 Brighton and Sussex University Hospital NHS Trust, Brighton, United Kingdom, 5 HIV i-Base, London, United Kingdom

* f.lampe@ucl.ac.uk
¶ Membership of for the AURAH2 Study Group is listed in the Acknowledgments.

## Abstract

The effectiveness of population-level intervention for HIV elimination is influenced by individual-level variation in sexual behaviour. We assess within-person changes in the frequency of condomless anal sex with two or more partners (CLS2+), estimate the transition probabilities and examine the predictors of transitions among a prospective cohort of HIV-negative gay, bisexual, and other men who have sex with men (GBMSM). Participants were recruited through one of three sexual health clinics in London and Brighton (July 2013 to April 2016) and self-completed a baseline paper questionnaire in the clinic. During follow-up, they were invited to complete four-monthly questionnaires twice a year and subsequent annual online questionnaires once a year (March 2015 to March 2018). We used Markov chain models to estimate transition probabilities from 'higher-risk' (CLS2+) to 'lower-risk' (no CLS2+) and vice versa, and to assess factors associated with transitions between different sexual risk levels. Among 1,162 men enrolled in the study, 622 (53.5%) completed at least one online questionnaire. Higher-risk behaviour was reported in 376/622 (60.4%) men during online follow-up. Overall, 1,665/3,277 (37.5%) baseline and follow-up questionnaires reported higher-risk behaviour. More than 60% of men (376/622) reported higher-risk behaviour at least one period during the follow-up, while 39.5% of men (246/622) never reported CLS2+ during the follow-up. In the next four months, the estimated probability of continuing higher-risk behaviour among men who reported higher-risk behaviour was 78%. Calendar time, recent HIV tests, PrEP and PEP use were the predictors of staying in higher-risk behaviour, while less stable housing status was associated with switching to lower-risk behaviour. Among men who reported lower-risk behaviour, the probability of engaging in the same behaviour was 88%. Recent HIV tests, PrEP and PEP use, recreational drugs, chemsex-associated drug and injection drugs, and bacterial

**Data availability statement:** Any personally identifiable data cannot be made publicly available as this study was conducted with approval from the National Research Ethics Service (NRES) committee, which only allows data from the studies to be released after the NRES provides written approval. This is in order to protect participants' privacy. A de-identified dataset sufficient to reproduce the study findings can be made available upon written request, after approval from the NRES committee. To submit a request for these data, please contact hampstead.rec@hra.nhs.uk or queries@hra.nhs.uk or go to https://www.hra.nhs.uk/about-us/contact-us/.

**Funding:** The AURAH2 study was funded by the National Institute for Health Research (NIHR) under its Programme Grants for Applied Research Programme (RP-PG-1212-20006). The views expressed in this study are those of the author(s) and not necessarily those of the NIHR, or the Department of Health and Social Care. The funder had no role in study design, data collection and anakysis, decision to publish, or preparation of the manuscript.

**Competing interests:** The authors have declared that no competing interests exist.

STIs diagnosis were the predictors of switching to higher-risk behaviour. Our results indicate that at any one point in time, the majority of GBMSM are at low risk for HIV acquisition, although many experience short periods in which they are at higher risk. Markers of transitions can be utilized to identify which GBMSM are likely to increase or decrease their risk, thus helping the timing of HIV prevention interventions.

## Introduction

HIV has become endemic in gay, bisexual, and other men who have sex with men (GBMSM) populations in many high-income settings, including the United Kingdom (UK), the United States (US), the Netherlands and Australia [1]. Data suggest that the effect of individual-level variation in sexual behaviour over time is crucial to understanding why HIV transmission rates have stayed high in this population and deserves further study of its complexity, as it plays an important role in driving the HIV epidemic [2–4]. Modelling studies have also shown that behavioural heterogeneity in terms of number of sexual partners [5] and short-term increases in risk behaviour (sometimes referred to as episodic risk) [6] have a significant impact on the chances of HIV elimination by test-and-treat strategies, confirming that individual-level sexual behaviour is a major determinant of population-level intervention efficacy. To reduce HIV transmission, preventative interventions such as the implementation of pre-exposure prophylaxis (PrEP) and other strategies must be timed perfectly. It is important to identify the behavioural windows associated with elevated HIV risk, and to promptly initiate and terminate such interventions for successful prevention and to prevent medication overuse. Understanding how long people stay in periods of high risk is important from a clinical perspective as well as to inform HIV models.

There are several approaches to identify individual-level changes using repeated observations of the same variable over time [7], including investigating whether there are multiple typical trajectories, leading to the characterisation of latent subgroups of individuals who share a typical profile over time [8]. Transition analyses, in which the probabilities of transitions among behaviour patterns over time are estimated, have also been applied to quantify changing behaviours over time and are useful to identify risk patterns in longitudinal data, characterise high-risk GBMSM and quantify individual transitions over time [9,10].

There are limited data on within-person changes in sexual behaviour among HIV-negative GBMSM, especially in the UK [11–22]. In this paper, we characterise longitudinal trajectories of sexual behaviour, the probabilities of transitions between sexual behaviour levels based on participants reported condomless anal sex with two or more partners (CLS2+) in a four-month period, and the predictors that affect the transitions among HIV-negative GBMSM participating in the AURAH2 study.

## Materials and methods

### Study design and participants

Methodological details of the study have been published previously [23]. The AURAH2 study was a prospective cohort study that recruited GBMSM who were HIV-negative or of unknown HIV status from three large sexual health clinics in London and Brighton (56 Dean Street, London; Mortimer Market Centre, London; Claude Nicol Clinic, Brighton) from July 2013 to April 2016. Participants were eligible if they were aged 18 years or older and had attended the study clinics for routine testing for sexually transmitted infections (STI) or HIV. Men were classified as GBMSM for the purposes of the analysis if they met at least one of the following criteria: (i) reported being

gay or bisexual, (ii) reported anal sex with a man in the past three months, or (iii) reported having disclosed to their family, friends or workmates as being gay, bisexual and/or attracted to men.

The AURAH2 study expanded on the design of the 'Attitudes to and Understanding Risk of Acquisition of HIV (AURAH)' questionnaire study, conducted between 2013 and 2014 [24]. The AURAH study design was cross-sectional recruited participants from 20 sexual health clinics in England. Data were collected using a self-completed paper questionnaire. Due to the transitions from the AURAH questionnaire study to the AURAH2 prospective study, participants were recruited into the AURAH2 study through two different routes, from the AURAH study (March–April 2015) and direct recruitment (November 2014–December 2016). Participants who joined the study from the AURAH study had already completed the baseline paper questionnaire since 2013 (July 2013–November 2014), and they were not asked to fill in a new baseline questionnaire. Participants who consented to the study through direct recruitment completed a confidential baseline paper questionnaire in the clinic (November 2014–April 2016).

During the follow-up period, participants self-completed subsequent four monthly and annual questionnaires that were available online from March 2015 until March 2018. Participants were invited to complete the four-monthly questionnaire twice a year, and subsequently to complete the more extensive annual questionnaire yearly. Given the three-year follow-up period, if a participant recruited in March 2015 completed every questionnaire that they were prompted to, they would complete a total of ten questionnaires over three years; one baseline paper-based in the clinic, nine online follow-up questionnaires, of which at least three were annual, and six were four-monthly questionnaires.

The baseline questionnaire gathered information on demographic, socioeconomic, lifestyle, health and wellbeing-related factors, knowledge and understanding of HIV, sexual behaviours, STI diagnoses, and PrEP and post-exposure prophylaxis (PEP) use. The four-monthly questionnaires assessed information on HIV status, HIV testing history, sexual behaviours, and lifestyle factors. Annual questionnaires captured the same information as the four-monthly questionnaire and additional information on PrEP and PEP use in the past year, relationship status, and health and wellbeing factors as assessed on the baseline questionnaire.

### Ethics approval and participant consent

All participants provided written, informed consent before taking part. Consent to participate in the study included consent for linkage to the UK Health Security Agency (UKHSA)'s datasets at the end of the study using limited participant identifiers. The AURAH2 study was approved by the designated research ethics committee, The National Research Ethics Service (NRES) committee London-Hampstead, ref: 14/LO/1881 in November 2014 [23], while the AURAH cross-sectional study was approved previously in April 2013 [24]. Based on the research protocol and all versions of study documents, the AURAH2 study subsequently received permission for clinical research at the three participating National Health Service (NHS) sites: Chelsea and Westminster NHS Foundation Trust, Central and North West London NHS Foundation Trust, and the Brighton and Sussex University Hospitals NHS Trust. The AURAH2 study was registered on the NIHR clinical research network portfolio. Authors had no access to information that could identify individual participants during or after data collection.

### Measures

**'Higher-risk' sexual behaviour measure.** Condomless anal sex with two or more partners (CLS2+) in the past three months was defined as the measure of 'higher-risk' sexual behaviour in this study. No condomless anal sex (CLS) or CLS with one partner only was defined as 'lower-risk' sexual behaviour.

**Socio-demographic, health and lifestyle, and other sexual/HIV-related behaviours measures.** We considered all variables collected in the study as possible predictors. Socio-demographic variables were age group, country of birth and ethnicity, sexual identity, university education status, ongoing relationship status, employment status, financial status, and housing status.

Health and lifestyle factors were recreational drug use, chemsex use, injection drug use, depressive symptoms (defined as a score of ≥10 on the Patient Health Questionnaire [PHQ-9], which is the standard cut-off score used to define clinically significant depressive symptoms) [25], anxiety symptoms (defined as a score of ≥10 on the Generalised Anxiety Disorder Scale [GAD-7], which represents the standard cut-off to define anxiety disorder) [26], and higher alcohol consumption (a score of ≥6 on a modified version the AUDIT-C WHO alcohol screening tool questionnaire, first two questions only) [27]. A total score of six was chosen given that AURAH2 participants were only asked the first two questions of the WHO AUDIT-C questionnaire rather than the full AUDIT-C.

Six other measures of sexual and other HIV-related behaviours were considered as predictors for analyses: CLS, group sex, bacterial STI diagnoses, recent HIV test, pre-exposure prophylaxis (PrEP), and post-exposure prophylaxis (PEP) use.

Country of birth and ethnicity, sexual identity, university education status, employment status, financial status, and housing status were fixed variables derived from the baseline questionnaire, whereas age, sexual/ HIV-related behaviours, and health and lifestyle factors were time-varying variables (using either four-monthly or annual questionnaires). The proportion of missing data in the AURAH2 data was minimal (≤5% in baseline data and 1% in online questionnaires). For specific binary variables, a non-response to a question was interpreted that a measure or event had not occurred. This was because there appeared to be a pattern in the responses to questions where only behaviours or symptoms experienced were ticked. As a result, missing values for variables were treated as 'No', except for variables: age, country of birth and ethnicity, sexual identity, financial status, and housing status, for which data on men with missing values were excluded. Initial analyses that were undertaken to investigate whether excluding missing values (when defining each variable) impacted findings, demonstrated that this was not the case.

## Statistical analysis

Analysis of within-person changes was done using data from men who completed at least an online questionnaire. First, the lasagna plotting framework is presented to graphically visualized changes in CLS2+ by plotting individual data as horizontal layers on top of each other. Each layer represents a participant, and each column represents a time point (visit or number of the questionnaire). This framework was also used to identify if a participant missed a questionnaire at any point. For this analysis, we classified responses at each questionnaire into four groups so that we could also assess the patterns of missingness: higher-risk, lower-risk, skipping a questionnaire (did not complete a questionnaire at a specific time, but later came back), and lost (never came back to the study). The plot shows the entire-row sorted (based on sexual behaviour reported) plot by plotting higher-risk behaviour at the bottom of the plot, and lower-risk behaviour at the top of the plot first (the first column), followed by skipping a questionnaire, and then lost (never came back to the study).

To estimate transition probabilities from 'higher-risk' to 'lower-risk' and vice versa, homogenous/ one component discrete-state Markov chain models were fitted. The mixmcm procedure in Stata [28], was used to investigate individuals' predictability over time. The mixmcm performs multiple multinomial logistic regressions to estimate parameters of transition probabilities (logits or log-odds for each cell of the transition table) [28]. Multinomial

regression values were modelled using longitudinal data, providing probabilities (probabilities being a function of logits). To account for the possibility of incomplete information within the data (that is, unobserved heterogeneity), the model was fitted with maximum likelihood using the expectation-maximization algorithm. To observe at least one transition, for this analysis, data from participants who completed at least an online questionnaire and had at least two time-points data of follow-up were used. Markov chain models were also fitted to assess the contributions of socio-demographic, health and lifestyle, and other HIV-related behaviours factors in explaining the odds of remaining in the initial state or switching to another state. Covariates were examined individually in univariable models. The resulting coefficients of the explanatory variables explain the degree to which covariates contribute to individual transitions, where remaining in the current behaviour is the reference scenario. In general, the exponentiated coefficients are odds ratios in multinomial logit models. Therefore, exponentiated coefficients are presented as odds ratios.

An additional cross-sectional analysis was also done using GEE logistic models to examine factors associated with reporting CLS2+ in the past three months, using all available baseline and follow-up questionnaires from 542 participants who had at least two-time points data of follow-up. This was done to compare results with those in the transition analysis using Markov chain models. Odds ratios (ORs) for factors, (i) unadjusted, (ii) adjusted for age, and (iii) adjusted for age, country of birth and ethnicity, sexual identity and university education status with their 95% CIs are presented. The adjustment for variables (ii) was determined a priori on factors that were considered not to be influenced by CLS2 + .

Lastly using data from all baseline, four-monthly, and annual follow-up questionnaires of every participant enrolled in AURAH2, we also calculated the prevalence of CLS2+ and CLS according to calendar period, using three-month periods from the first enrolment (July 2013) to the end of the AURAH2 study period (March 2018). Univariable generalised estimation equation (GEE) models with a logit link and robust standard errors (SEs) were used to assess trends over calendar time in prevalence over the period from July 2013 to March 2018, accounting for multiple questionnaire responses from individual participants. The calendar year was fitted as a continuous variable to obtain a test for linear trend.

All analyses were done using Stata SE version 15.1. The lasagna plot was reproduced using Tableau 2018.3.

## Results

### Characteristics of the participants

Between July 2013 and April 2016, a total of 1,162 HIV-negative GBMSM completed a baseline questionnaire and were enrolled in the study (**Table 1**). At baseline, the mean age of participants was 34 years (interquartile range [IQR]: 26–39), 81.9% were of White ethnicity, 93.6% self-reported being gay, 74.4% had a university degree, 82.9% reported being employed, and 77.4% always had money to cover basic needs. 12.2% of men reported clinically significant depressive symptoms, 9.3% reported clinically significant anxiety symptoms, and 13.0% reported higher alcohol consumption. The proportion of missing responses was low.

Of the 1,162 men enrolled, 622 (53.5%) of them completed at least one online follow-up questionnaire, and 411 (35.4%) engaged in the follow-up within the final six months of the study. When comparing with 540 men who completed only the baseline questionnaire, the 622 men who continued on the study were older (median age 33 versus 30 years, *p-value* from chi-square test < 0.001), had greater financial security (had money all of the time 82.4% versus 71.1%, $p < 0.001$), had more stable housing (homeowner 33.0% versus 21.0%, $p < 0.001$), were

**Table 1. Baseline socio-demographic and health and lifestyle characteristics among participants who completed the baseline and online follow-up questionnaire in the AURAH2 study, 2013–2018.**

| Socio-demographic and health and lifestyle characteristics | All men enrolled in the study (N=1162) | | Completing only baseline questionnaire (n=540) | | Completing baseline and at least an online question-naire (n=622) | | p-value* |
|---|---|---|---|---|---|---|---|
| | N | n (%) | N | n (%) | N | n (%) | |
| **Age category** | 1153 | | 538 | | 615 | | **<0.001** |
| <25 | | 275 (23·9%) | | 142 (26.4%) | | 133 (21.6%) | |
| 25–29 | | 207 (17·9%) | | 120 (22.3%) | | 87 (14.1%) | |
| 30–34 | | 227 (19·6%) | | 104 (19.3%) | | 123 (20.0%) | |
| 35–39 | | 156 (13·6%) | | 68 (12.6%) | | 88 (14.3%) | |
| 40–44 | | 121 (10·5%) | | 52 (9.7%) | | 69 (11.2%) | |
| ≥45 | | 167 (14·5%) | | 52 (9.7%) | | 115 (18.7%) | |
| **Median age, years (IQR)** | **31 (26–39)** | | **30 (25–37)** | | **33 (26–41)** | | |
| **Country of birth & ethnicity** | 1150 | | 538 | | 612 | | 0.205 |
| Born in the UK, White | | 568 (49·4%) | | 250 (46.5%) | | 318 (51.9%) | |
| Born in the UK, Other ethnicity$ | | 60 (5·2%) | | 31 (5.8%) | | 29 (4.7%) | |
| Non-UK born, White | | 374 (32·5%) | | 179 (33.3%) | | 195 (31.9%) | |
| Non-UK born, Other ethnicity | | 148 (12·9%) | | 78 (14.5%) | | 70 (11.4%) | |
| **Sexual identity** | 1150 | | 535 | | 615 | | 0.179 |
| Gay | | 1076 (93·6%) | | 495 (92.5%) | | 581 (94.6%) | |
| Bisexual/ other | | 74 (6·4%) | | 40 (7.5%) | | 33 (5.4%) | |
| **University education status** | 1146 | | 540 | | 617 | | **0.017** |
| Yes | | 853 (74·4%) | | 381 (70.5%) | | 474 (76.8%) | |
| No | | 293 (25·6%) | | 159 (29.5%) | | 143 (23.2%) | |
| **Employment status†** | 1149 | | 540 | | 617 | | **0.010** |
| Employed | | 952 (82·9%) | | 503 (93.1%) | | 548 (88.8%) | |
| Unemployed/ other | | 197 (17·1%) | | 37 (6.9%) | | 69 (11.2%) | |
| **Financial status**** | 1158 | | 540 | | 618 | | **<0.001** |
| All of the time | | 896 (77·4%) | | 384 (71.1%) | | 509 (82.4%) | |
| Most of the time | | 194 (16·8%) | | 114 (21.1%) | | 81 (13.1%) | |
| Sometimes/ No | | 68 (5·8%) | | 42 (7.8%) | | 28 (4.5%) | |
| **Housing status‡** | 1147 | | 539 | | 608 | | **<0.001** |
| Homeowner | | 314 (27·4%) | | 113 (21.0%) | | 201 (33.0%) | |
| Renting | | 680 (59·3%) | | 351 (65.1%) | | 328 (54.0%) | |
| Unstable/ other | | 153 (13·3%) | | 75 (13.9%) | | 79 (13.0%) | |
| **Ongoing relationship status** | 1159 | | 540 | | 619 | | 0.169 |
| Yes | | 465 (40·2%) | | 204 (37.8%) | | 257 (41.5%) | |
| No | | 693 (59·8%) | | 336 (62.2%) | | 362 (58.5%) | |
| **Higher risk alcohol consumption (WHO AUDIT-C ≥6)** | 1159 | | 540 | | 619 | | 0.941 |
| No | | 1008 (87·0%) | | 469 (86.8%) | | 537 (86.8%) | |
| Yes | | 151 (13·0%) | | 71 (13.2%) | | 82 (13.2%) | |
| **Depressive symptoms (PHQ-9 score ≥10)** | 1159 | | 540 | | 619 | | 0.991 |
| No | | 1018 (87·8%) | | 474 (87.8%) | | 544 (87.9%) | |
| Yes | | 141 (12·2%) | | 66 (12.2%) | | 75 (12.1%) | |
| **Anxiety symptoms (GAD7 score ≥10)** | 1159 | | 540 | | 619 | | 0.057 |
| No | | 1033 (89·1%) | | 471 (87.2%) | | 562 (90.8%) | |
| Yes | | 126 (10·9%) | | 69 (12.8%) | | 57 (9·2%) | |

* p-value from χ2 test for differences between men who completed only baseline questionnaire (n = 540) and men who continued on the study by completing at least an online questionnaire (n = 622).

*(Continued)*

**Table 1.** (Continued)

\*\* Having enough money to cover basic needs, e.g., for food and heating.

§ Other ethnicity includes Black, Asian, Mixed, and other ethnic group.

† Renting housing includes private renting and renting from council or housing association; unstable or other housing includes temporary accommodation, staying with friends or family, other accommodation, and homeless.

‡ Employed group includes full-time (n = 845) and part-time (n = 107) employment/ self-employment; Unemployed/ other group includes unemployed registered or not registered for benefits (n = 60), sick or disabled (n = 6), retired (n = 24), and other (student or training or looking after home or dependents or other) (n = 107)

SD: standard deviation, IQR: interquartile range, AUDIT-C: alcohol use disorders identification test-consumption, PHQ-9: patient health questionnaire-9, GAD-7: generalised anxiety disorder assessment-7

more likely to have university-level education (76.8% versus 70.5%, $p = 0.017$) and less likely to be employed (88.8% versus 93.1%, $p = 0.010$) (Table 1, third and last columns).

The number of follow-up questionnaires (four monthly and annual) completed by the 622 men who completed at least one online questionnaire by the end of the study period was 3,277. Participants completed a median of 6 (IQR: 3–7) online questionnaires.

### Within-person changes in the frequency of CLS2+

The lasagna plot of the AURAH2 study among 622 men is shown in Fig 1. Lower-risk behaviour (green surfaces) covered the majority of the plot. A consistent trend was observed in the reporting of sexual behaviours, together with an increasing frequency of loss to follow-up and a decreasing frequency of skipping a questionnaire over time. The majority of men (>60%) reported higher-risk behaviour at least one period during the follow-up, while 39.5% of men never reported CLS2+ (either consistently low risk since the first up to the 9th online questionnaire or changed to skip/lost). 16.7% men never reported lower-risk behaviour (either had CLS2+ at every questionnaire or changed to skip/lost).

**We illustrate** the proportion of changes from a different initial state (*higher-risk; lower-risk; skipping questionnaire; loss to follow-up*) within an individual in each questionnaire in Appendix 1.

### Overall transition probabilities

The transition probabilities were calculated using data among the 542 men who completed at least two consecutive questionnaires during follow-up (**Table 2**). The estimated probability of transitioning from having CLS2+ to not reporting having CLS2+ in the next period is 0.22, the probability of staying in higher-risk behaviour is therefore **0.78**, which implies that continuing having CLS2+ is quite persistent behaviour. Similarly, the probability of continuing reporting not having CLS2+ is **0.88**, which exhibits a high probability to remain in the same behaviour for two consecutive questionnaires.

### Predictors of transition probabilities

*Explanatory variables of transition probabilities from 'higher-risk' to 'lower-risk' behaviour.* **Table 3** presents the exponentiated coefficients (odds ratios) of the explanatory variables of transition probabilities. Recent HIV test, PrEP use, PEP use, and calendar year were associated with reduced probability of transition from having CLS2+ to have one or none CLS partner, while less stable housing status increased the probability. In other words, men who reported an HIV test in the past three months (OR 0.38, 95% CI 0.27–0.54, $p < 0.001$), the use of PrEP in the past 12 months (OR 0.13, 95% CI 0.05–0.35, $p < 0.001$), and the use of PEP in the past 12 months (OR 0.31, 95% CI 0.10–0.91, $p = 0.033$) were less likely to transition to safer behaviour and more likely to remain in higher-risk behaviour

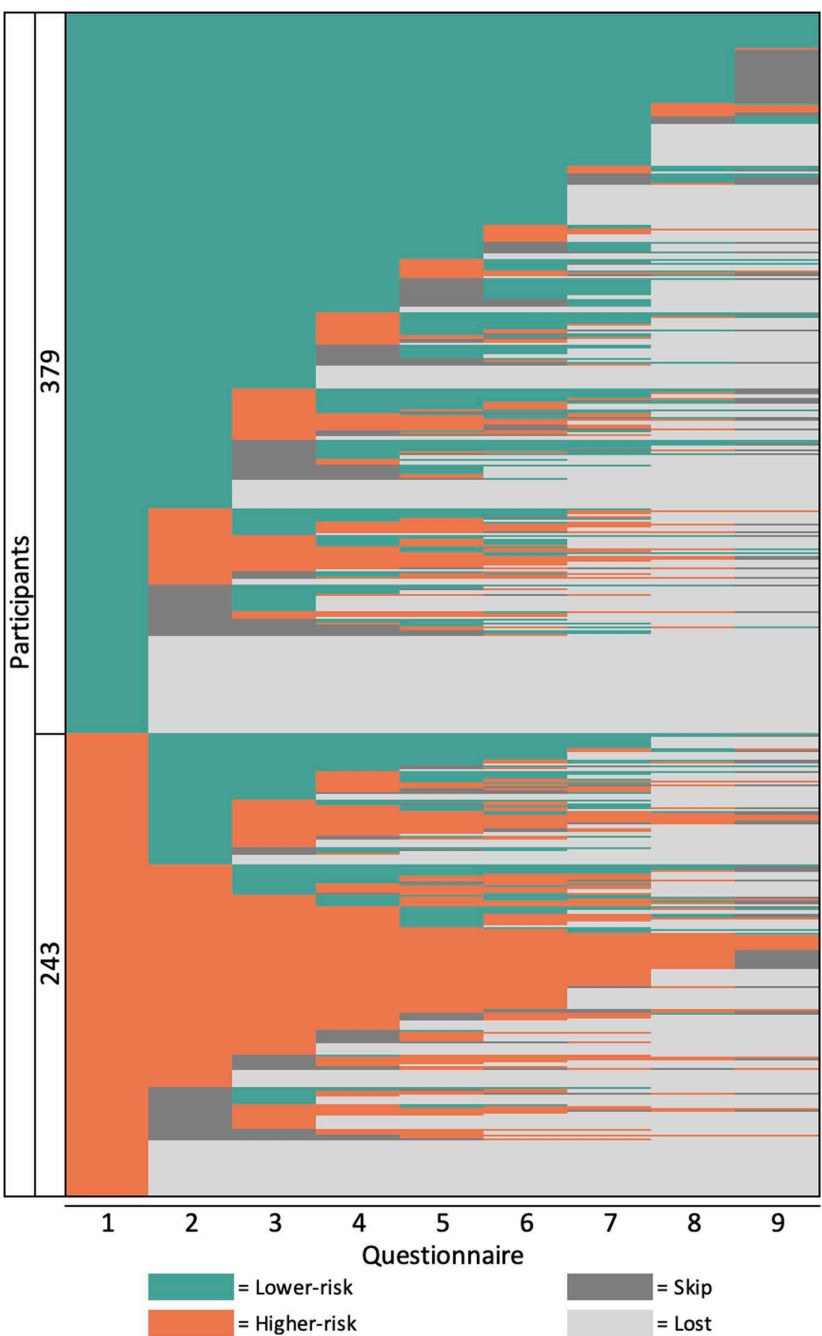

**Fig 1. Entire row sorted lasagna plot illustrating trajectories in reporting sexual behaviours among 622 GBMSM in the AURAH2 study (n = 5,598 observations)＊.** ＊ GBMSM: gay, bisexual, and other men who have sex with men.

in subsequent questionnaires. More recent calendar year (OR 0.78, 95% CI 0.63–0.96, $p = 0.018$) also decreased the probability of men transitioning to safer risk behaviour. On the other hand, men with less stable housing tended to transition to safer risk behaviour than to remain in higher-risk behaviour in the subsequent questionnaire (OR 1.04, 95% CI 1.01–1.07, $p = 0.016$).

**Table 2. Transition probabilities for sexual risk behaviour among 542 GBMSM in the AURAH2 study.**

| State | transition probabilities | 95% CI |
|---|---|---|
| higher-risk to lower-risk | 0.22 | 0.19–0.24 |
| lower-risk to higher-risk | 0.12 | 0.11–0.13 |

CI, confidence interval.

**Table 3. Explanatory variables of transition probabilities from univariable Markov transition probability models among 542 GBMSM in the AURAH2 study, 2015–2018.**

| | Transition probabilities from *'higher-risk'* to *'lower-risk'* | | Transition probabilities from 'lower-risk' to 'higher-risk' | |
|---|---|---|---|---|
| Explanatory covariates | OR (95% CI) | p-value | OR (95% CI) | p-value |
| **Demographics characteristics** | | | | |
| Age | 0.98 (0.97–1.00) | 0.056 | 1.00 (0.99–1.02) | 0.492 |
| Country of birth and ethnicity | 1.05 (0.92–1.20) | 0.482 | 1.05 (0.92–1.19) | 0.481 |
| Sexual identity | 0.57 (0.27–1.22) | 0.154 | 0.88 (0.49–1.59) | 0.680 |
| **Socio-economic characteristics and partnership status** | | | | |
| University education status | 1.24 (0.88–1.76) | 0.216 | 0.87 (0.63–1.21) | 0.421 |
| Employed | 1.17 (0.72–1.89) | 0.526 | 1.24 (0.77–2.00) | 0.380 |
| Financial status | 0.75 (0.53–1.05) | 0.091 | 1.01 (0.76–1.33) | 0.949 |
| Housing status | **1.04 (1.01–1.07)** | **0.016** | **0.97 (0.94–0.99)** | **0.035** |
| Ongoing relationship | 0.96 (0.71–1.30) | 0.807 | 1.12 (0.84–1.48 | 0.432 |
| **Other HIV-related behaviour characteristics** | | | | |
| Recent HIV test | **0.38 (0.27–0.54)** | **<0.001** | **1.80 (1.33–2.43)** | **<0.001** |
| PrEP use | **0.13 (0.05–0.35)** | **<0.001** | **2.57 (1.13–5.86)** | **0.024** |
| PEP use | **0.31 (0.10–0.91)** | **0.033** | **2.54 (1.23–5.25)** | **0.012** |
| Bacterial STI diagnosis | 0.80 (0.57–1.14) | 0.225 | **2.12 (1.37–3.27)** | **0.001** |
| **Health and lifestyle characteristics** | | | | |
| Recreational drug use | 0.68 (0.37–1.27) | 0.229 | **2.02 (2.01–3.39)** | **0.009** |
| Injection drug use | 0.27 (0.03–2.12) | 0.214 | **5.82 (4.48–7.55)** | **<0.001** |
| Chemsex use | 0.76 (0.56–1.04) | 0.087 | **2.27 (1.60–3.21)** | **<0.001** |
| Higher risk alcohol consumption | 1.01 (0.65–1.54) | 0.999 | 1.04 (0.69–1.58) | 0.839 |
| Depressive symptoms | 0.57 (0.32–1.02) | 0.061 | 0.81 (0.49–1.34) | 0.414 |
| Anxiety symptoms | 0.68 (0.32–1.44) | 0.315 | 0.68 (0.38–1.24) | 0.210 |
| Calendar year | **0.78 (0.63–0.96)** | **0.018** | 1.12 (0.93–1.34) | 0.231 |

CI, confidence interval; OR, odds ratio; PrEP, pre-exposure prophylaxis; PEP, post-exposure prophylaxis; STI, sexually transmitted infection.

**Explanatory variables of transition probabilities from 'lower-risk' to 'higher-risk' behaviour.** Less stable housing status decreased the probability of men to transition out from the lower-risk behaviour in a subsequent questionnaire (OR 0.97, 95%CI 0.94–0.99, $p = 0.035$), while reporting a recent HIV test (OR 1.80, 95% CI 1.33–2.43, $p < 0.001$), PrEP use (OR 2.57, 95% CI 1.13–5.86, $p = 0.024$), and PEP use (OR 2.54, 95% CI 1.23–5.25, $p = 0.012$) predicted the transition from lower-risk to higher-risk behaviour. In addition, the probabilities of switching to CLS2+ were predicted by a bacterial STI diagnosis (OR 2.12, 95% CI 1.37–3.27, $p = 0.001$), the use of recreational drugs (OR 2.02, 95% CI 2.01–3.39, $p = 0.009$), injection drugs (OR 5.82, 95% CI 4.48–7.55, $p < 0.001$), and chemsex drugs (OR 2.27, 95% CI 1.60–3.21, $p < 0.001$). The

use of injection drugs had the highest effect in predicting a switch to higher-risk behaviour, up to almost six-fold higher odds.

**Cross-sectional analysis of factors associated with condomless anal sex with two or more partners.** In age-adjusted models (Table 4), CLS2+ was more likely among men without university education (aOR 1.46, 95% CI 1.06–2.20, $p < 0.021$), compared to those with university education. Behavioural factors associated with reporting CLS2+ were similar to those explanatory factors of transition probabilities from lower-risk to higher-risk in the previous analysis, which were a recent HIV test, recreational drug use, injection drug use, chemsex use, the use of PrEP and PEP in the previous 12 months, and bacterial STI diagnosis. Calendar year as a continuous variable was also strongly associated with an increased odds of reporting CLS2+ (aOR 1.08, 95% CI 1.00–1.16, $p < 0.040$). The strongest association was found between CLS2+ and injection drug use (aOR 8.08, 95% CI 3.6–18.11, $p < 0.001$). Adjustment for age, country of birth and ethnicity, sexual identity and university education status produced similar results as the age-adjusted models.

### Trends in CLS2+ and CLS over follow-up period

Fig 2 shows trends in reporting CLS2+ and CLS in the past three months from 2013 to 2018. Overall, CLS2+ and CLS were reported in 1,665 and 2,973 baseline and follow-up questionnaires (37.5% and 66.9% of 4,338 total questionnaires). Prevalence of both sexual behaviours increased until the end of the follow-up period; CLS rose from 64.3% in the last two quarters of 2013 to 74.4% by January to March 2018 (unadjusted OR 1.09 per year, 95% CI 1.03–1.17, *p-value for linear trend from GEE-logistic model = 0.007*); CLS2+ increased from 46.4% in the period of July to December 2013, to 46.9% in January to March 2018 (unadjusted OR 1.09 per year, 95% CI 1.02–1.16, *p-trend = 0.010*). Both significant trends seem to be driven by data from 2015 onwards, as early data were relatively small numbers (*p-values* for 2015 onwards 0.001 (CLS2+) and 0.002 (CLS)).

### Discussion

Using data from a prospectively followed cohort of initially HIV-negative GBMSM in London and Brighton, we showed that while individuals comprising the membership of lower-risk or higher-risk behaviour categories changed somewhat from questionnaire to questionnaire, the overall proportions of each sexual risk group were generally stable during follow-up. There were moderate transitions across risk classes, regardless of the initial state; men had an overall 78% probability of continuing higher-risk behaviour and an 88% probability of remaining in the lower-risk behaviour in the subsequent questionnaires. Our study also demonstrated that while GBMSM vary in their level of risk over time, there remains a predominance of lower-risk behaviour. We also observed a slight increase in the trends of CLS2+ and CLS over time, which has been previously reported among all men enrolled in AURAH2 [29], and is consistent with findings from other UK and European studies [19–22,30].

Findings from limited previous research characterising transitions and individual trajectories in risk behaviour among HIV-negative GBMSM have been mixed. The Victorian Primary Care Network for Sentinel Surveillance on Bloodborne Viruses and STIs (VPCNSS) in Australia [9], the Amsterdam Cohort Studies (ACS) in the Netherlands [31], and the Multicenter AIDS Cohort Study (MACS) in the United States (US) [11] reported overall similar results with those from our analysis, that HIV-negative men exhibit relatively stable yet distinct patterns of sexual risk behaviour over time, and that the majority of GBMSM showed little behaviour change from one period to the next [9,11,31]. In the ACS, based on sexual behaviour score, risk levels were classified into three: low (73% of visits), medium (22%), and

**Table 4. Factors associated with reporting condomless sex with two or more partners from GEE logistic models among 542 GBMSM in the AURAH2 study***.

| | Unadjusted OR (95% CI) | p-value | Age-adjusted OR (95% CI) | p-value | Adjusted OR** (95% CI) | p-value |
|---|---|---|---|---|---|---|
| **Demographic characteristics** | | | | | | |
| **Age category, years** | 3131 obs | 0·174 | 3131 obs | 0·174 | 3121 obs | 0·118 |
| <25 | 1 (Ref) | | 1 (Ref) | | 1 (Ref) | |
| 25–29 | 0·91 (0·54–1·53) | | 0·91 (0·54–1·53) | | 0·94 (0·56–1·59) | |
| 30–34 | 1·14 (0·74–1·75) | | 1·14 (0·74–1·75) | | 1·18 (0·74–1·78) | |
| 35–39 | 1·40 (0·88–2·23) | | 1·40 (0·88–2·23) | | 1·34 (0·84–2·15) | |
| 40–44 | 1·68 (1·01–2·80) | | 1·68 (1·01–2·80) | | 1·60 (0·95–2·70) | |
| ≥45 | 1·46 (0·94–2·20) | | 1·46 (0·94–2·20) | | 1·39 (0·90–2·16) | |
| **Country of birth & ethnicity** | 3131 obs | 0·279 | 3131 obs | 0·699 | 3121 obs | 0·490 |
| Born in the UK, White | 1 (Ref) | | 1 (Ref) | | 1 (Ref) | |
| Born in the UK, Other ethnicity§ | 1·71 (0·97–3·02) | | 1·87 (1·06–3·33) | | 1·91 (1·07–3·39) | |
| Non-UK born, White | 0·98 (0·71–1·34) | | 0·99 (0·72–1·37) | | 1·01 (0·73–1·39) | |
| Non-UK born, Other ethnicity | 1·12 (0·72–1·74) | | 1·17 (0·75–1·82) | | 1·28 (0·82–2·00) | |
| **Sexual identity** | 3137 obs | 0·219 | 3121 obs | 0·290 | 3121 obs | 0·288 |
| Bisexual/ other | 1 (Ref) | | 1 (Ref) | | 1 (Ref) | |
| Gay | 1·51 (0·78–2·93) | | 1·42 (0·74–2·72) | | 1·46 (1·06–2·04) | |
| **Socio-economic characteristics and partnership status** | | | | | | |
| **University education status** | 3131 obs | **0·012** | 3131 obs | **0·021** | 3121 obs | **0·021** |
| Yes | 1 (Ref) | | 1 (Ref) | | 1 (Ref) | |
| No | **1·50 (1·09–2·06)** | | **1·46 (1·06–2·02)** | | **1·17 (0·96–1·43)** | |
| **Employment status** | 3147 obs | 0·639 | 3131 obs | 0·465 | 3121 obs | 0·457 |
| Employed | 1 (Ref) | | 1 (Ref) | | 1 (Ref) | |
| Unemployed/ other | 1·12 (0·63–1·09) | | 0·82 (0·49–1·38) | | 0·82 (0·49–1·38) | |
| **Financial status^** | 3147 obs | 0·231 | 3131 obs | 0·694 | 3121 obs | 0·931 |
| All of the time | 1 (Ref) | | 1 (Ref) | | 1 (Ref) | |
| Most of the time | 1·39 (0·92–2·12) | | 1·47 (0·96–2·23) | | 1·38 (0·90–2·11) | |
| Sometimes/ No | 0·79 (0·37–1·71) | | 0·72 (0·32–1·62) | | 0·68 (0·30–1·53) | |
| **Housing status** | 3103 obs | 0·049 | 3103 obs | 0·121 | 3093 obs | 0·055 |
| Homeowner | 1 (Ref) | | 1 (Ref) | | 1 (Ref) | |
| Renting | **0·70 (0·52–0·96)** | | 0·77 (0·54–1·10) | | 0·70 (0·48–1·01) | |
| Unstable/ other | 0·65 (0·41–1·02) | | 0·72 (0·43–1·19) | | 0·72 (0·43–1·18) | |
| **Ongoing relationship** | 3147 obs | 0·974 | 3131 obs | 0·642 | 3121 obs | 0·662 |
| Yes | 1 (Ref) | | 1 (Ref) | | 1 (Ref) | |
| No | 0·99 (0·75–1·32) | | 1·07 (0·80–1·43) | | 1·07 (0·80–1·42) | |
| **Other HIV-related behaviour characteristics** | | | | | | |
| **Recent HIV test in the past 3 months** | 2984 obs | **<0.001** | 2957 obs | **<0.001** | 2947 obs | **<0.001** |
| No | 1 (Ref) | | 1 (Ref) | | 1 (Ref) | |
| Yes | **1·50 (1·31–1·72)** | | **1·53 (1·33–1·76)** | | **1·54 (1·33–1·77)** | |
| **PEP use in the past 12 months** | 902 obs | **0.038** | 893 obs | **0.026** | 891 obs | **0.036** |
| No | 1 (Ref) | | 1 (Ref) | | 1 (Ref) | |
| Yes | **1·42 (1·02 –1·97)** | | **1·47 (1·05–2·06)** | | **1·44 (1·02–2·02)** | |
| **PrEP use in the past 12 months** | 902 obs | **<0.001** | 893 obs | **<0.001** | 891 obs | **<0.001** |
| No | 1 (Ref) | | 1 (Ref) | | 1 (Ref) | |
| Yes | **6·83 (4·69–9·96)** | | **7·10 (4·86–10·38)** | | **6·96 (4·73–10·25)** | |
| **Health and lifestyle characteristics** | | | | | | |

*(Continued)*

**Table 4.** (Continued)

| | Unadjusted OR (95% CI) | p-value | Age-adjusted OR (95% CI) | p-value | Adjusted OR** (95% CI) | p-value |
|---|---|---|---|---|---|---|
| **Recreational drug use in the past 3 months** | 926 obs | **<0.001** | 915 obs | **<0.001** | 913 obs | **<0.001** |
| No | 1 (Ref) | | 1 (Ref) | | 1 (Ref) | |
| Yes | **2·17 (1·59–2·98)** | | **2·35 (1·70–3·24)** | | **2·39 (1·72–3·33)** | |
| **Injection drug use in the past 3 months** | 922 obs | **<0.001** | 911 obs | **<0.001** | 909 obs | **<0.001** |
| No | 1 (Ref) | | 1 (Ref) | | 1 (Ref) | |
| Yes | **8·04 (3·62–17·83)** | | **8·08 (3·60–18·11)** | | **7·07 (3·39–14·70)** | |
| **Chemsex use in the past 3 months** | 3164 obs | **<0.001** | 3131 obs | **<0.001** | 3121 obs | **<0.001** |
| No | 1 (Ref) | | 1 (Ref) | | 1 (Ref) | |
| Yes | **3·38 (2·69–4·24)** | | **3·39 (2·70–4·26)** | | **3·35 (2·67–4·20)** | |
| **Bacterial STI diagnoses in the past 3 months** | 3153 obs | **<0.001** | 3121 obs | **<0.001** | 3111 obs | **<0.001** |
| No | 1 (Ref) | | 1 (Ref) | | 1 (Ref) | |
| Yes | **1·56 (1·33–1·82)** | | **1·59 (1·36–1·87)** | | **1·59 (1·35–1·87)** | |
| **Higher risk alcohol consumption (modified WHO AUDIT-C ≥6)** | 3147 obs | 0·825 | 3131 obs | 0·673 | 3121 obs | 0·705 |
| No | 1 (Ref) | | 1 (Ref) | | 1 (Ref) | |
| Yes | 0·95 (0·62–1·46) | | 0·91 (0·59–1·40) | | 0·92 (0·59–1·41) | |
| **Depressive symptoms (PHQ-9 score ≥10)** | 3147 obs | 0·708 | 3131 obs | 0·696 | 3121 obs | 0·794 |
| No | 1 (Ref) | | 1 (Ref) | | 1 (Ref) | |
| Yes | 1·09 (0·70–1·68) | | 1·09 (0·69–1·71) | | 1·06 (0·68–1·64) | |
| **Anxiety symptoms (GAD7 score ≥10)** | 3147 obs | 0·506 | 3131 obs | 0·662 | 3121 obs | 0·627 |
| No | 1 (Ref) | | 1 (Ref) | | 1 (Ref) | |
| Yes | 0·84 (0·49–1·41) | | 0·89 (0·52–1·50) | | 0·88 (0·52–1·48) | |
| **Calendar year** | 3164 obs | **0·031** | 3131 obs | **0·040** | 3121 obs | **0·046** |
| | **1·08 (1·01–1·16)** | | **1·08 (1·00–1·16)** | | **1·07 (1·00–1·15)** | |

*Ethnicity, sexual identity, education, employment, money status, and housing status are fixed variables; Age, lifestyle characteristics, and HIV-risk behaviour are time-updated variables.

**Adjusted for age, country of born and ethnicity, sexuality and education level

^ Having enough money to cover basic needs, e.g., for food and heating

OR: odds ratio, CI: confidence interval, PrEP: pre-exposure prophylaxis, PEP: post-exposure prophylaxis, STI: sexually transmitted infection, AUDIT-C: alcohol use disorders identification test-consumption, PHQ-9: patient health questionnaire-9, GAD-7: generalised anxiety disorder assessment-7

high risk (5%) [31]. For GBMSM at low risk, the six-month probability of increasing risk was 11%. For GBMSM at medium risk, the probability of increasing to high risk was 8%, while the probability of decreasing to low risk was 33%. For GBMSM at high risk, the probability of decreasing risk was 43%. The slight difference observed in the probability of transition to and out from high risk level to other levels was probably due to the difference in the categorisation of risk levels.

Unlike in AURAH2, data from the Young Men's Health Study [13] and the CDC Collaborative Seroincidence Study in the US [3,4] observed little stability in sex risk from one wave to the next and a high level of movement between dichotomous sexual roles (insertive/ receptive, with/without condom, anal/oral, and HIV status of partners). Likely the differences were due to the difference in characteristics of men included in the cohort (younger men in the Young Men's Health Study; mean age 22 years) and the use of early data in CSS (data from 1992–1996) [3,4,13].

As in AURAH2, recreational drug use, including chemsex, has been consistently linked to CLS measures in studies of GBMSM in the UK and other high-income countries [32–39].

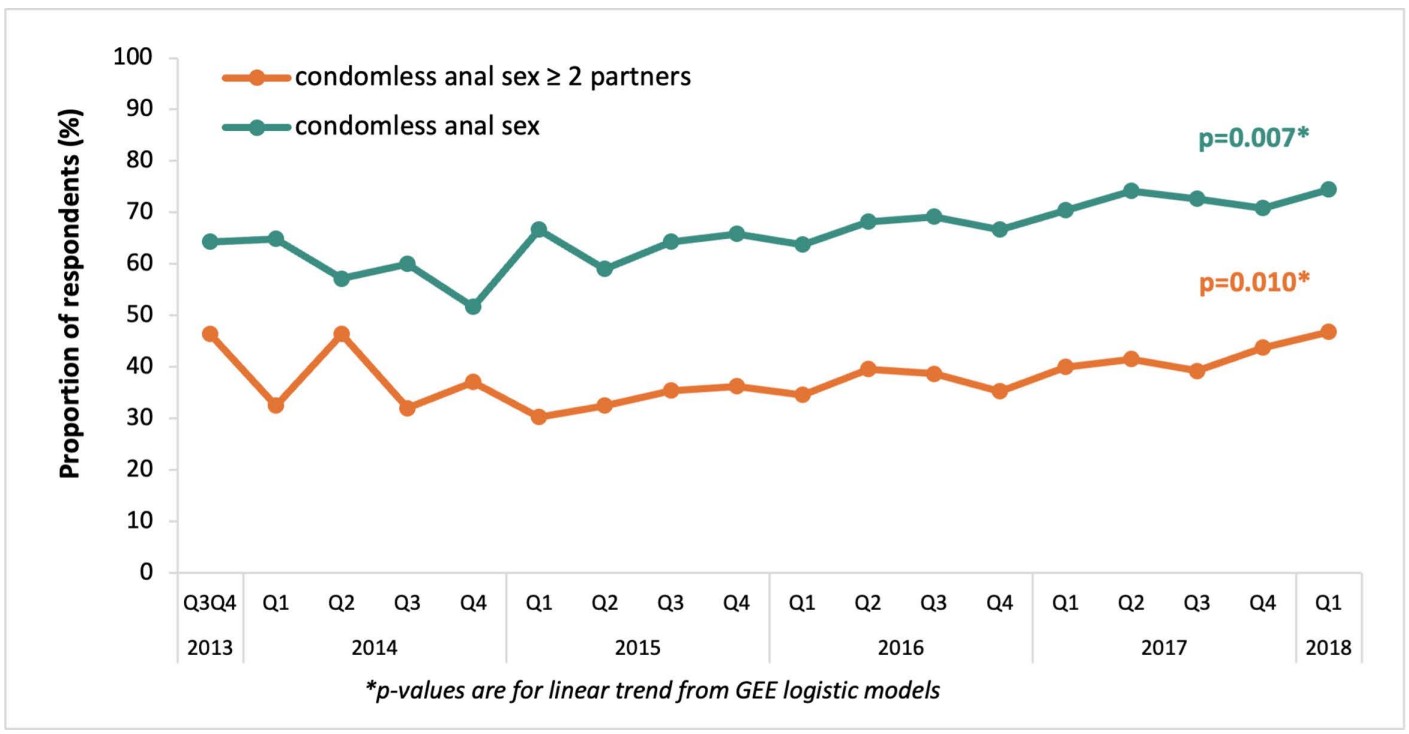

**Fig 2. Prevalence of CLS and CLS with two or more partners in the past three months over time among GBMSM in the AURAH2 study, 2013–2018\*.** *Data from baseline, four-monthly, and annual questionnaires, missing values on CLS and CLS with two or more partners were treated as No (N = 1,162 participants provided 4,338 questionnaires; one questionnaire was excluded from the analysis due to missing.

Associations between sexual behaviour trajectories and transitions between sexual risk class and substance use are also in line with previous data from the MACS, the ACS, and the Young Men's Health study [11,13,31].

There was a lack of association between age and CLS2+ in AURAH2, while evidence from previous studies suggests that younger men are more likely to engage in CLS, including with multiple partners and partners of an unknown or sero-different HIV status [11,17,22,40–42]. It is possible that in AURAH2, the age associations were affected by other socio-demographic characteristics of men attending sexual health clinics. In terms of other factors, findings from the AURAH2 study are similar to other UK studies of sexual behaviour among GBMSM, in which there was some evidence to suggest that ethnicity and sexual identity were not associated with multiple CLS partners and CLS partners of an unknown/sero-different HIV status [43,44]. We also found a protective effect of less stable housing status against CLS with two or more partners. One possible explanation is that homeless men or men with less stable housing status have more barriers and difficulties to maintain stable relationships or that they have less frequent sexual contact. In AURAH2, the use of PrEP was also associated with continuing higher-risk behaviour. The use of PrEP might have induced risk compensation, defined as increased sexual risk behaviour, which might be leading to increased diagnosis of bacterial STIs.

The impact of behavioural changes on intervention effectiveness has received less attention in the literature, owing to the necessity for high-quality longitudinal data to describe behaviour change. While identifying transitions and trajectories in sexual behaviour is important, whether these short-term (transitions between sexual behaviour state) and

long-term (trajectories of sexual behaviour over time) changes may undermine or enhance the impact of HIV prevention is still unclear. Modelling studies have shown that PrEP and 'test and treat' strategies could enormously impact the HIV epidemic among GBMSM if sexual risk behaviour does not change [5,45]. Therefore, if there is evidence of heterogeneity in sexual behaviour within-individual among GBMSM over time as in our study, future studies can focus on using mathematical modelling to explore how the individual transitions affect the reduction in HIV prevalence or incidence achieved by HIV intervention programmes compared to a population with constant risk levels. It would be informative to research or update 'static' models that do not include sexual trajectories with the addition of sexual trajectories, such as the impact of PrEP on HIV prevalence or reduction of HIV transmission by integrating sexual trajectories or transitions (such as how many men will transition in or out of high risk) and which intervention would be more cost-effective. Furthermore, there is the need to build models of PrEP use that can support people alternating between periods of use and non-use, as well as switching between dose, regimens or how the PrEP is administered as they become available and approved for use in the UK or England. This could also apply to the combination of different types of PrEP.

All of the markers of transitions to higher-risk behaviour discovered in this study might be used to identify GBMSM who are likely to increase their risk in the next four months and these men might benefit from PrEP as a tool that can be used for that short period of the enhanced risk. Likewise, factor linked with switching to lower-risk among GBMSM at higher risk, such as less stable housing status, may be utilized to decide whether a person should stop using PrEP or switch to an event-driven schedule. This will prevent excessive expenses or drugs overuse. Determinants linked to risk increase or decrease may be useful in identifying GBMSM who are likely to modify their behavior quickly, allowing for better scheduling of interventions. For instance, optimizing STI/HIV testing intervals (e.g., after four months), focusing behavior change treatments on GBMSM who are likely to increase risk, or starting or stopping PrEP use on time. This information might be used by nurses and clinicians so that they can ensure these men are aware of all the prevention tools at their disposal.

CLS2+ was considered as the measure of 'higher-risk' behaviour in this study, while it is possible that this does not reflect 'true' high-risk sexual behaviour. Within the literature there is a lack of consensus about the definition of high-risk sexual behaviour [46–48]. Among the measures of sexual behaviour in AURAH2, CLS with one partner was considered to be less likely to represent behaviour potentially linked to high-risk of acquisition of HIV because this may occur in the context of a long-term relationship where HIV status is known with more confidence. CLS with an HIV-positive partner when the HIV-positive partner is taking antiretroviral drugs with good adherence and has a fully suppressed viral load is also not a high-risk behaviour because the risk of HIV transmission through anal sex is zero in this context [49–51]. Therefore, CLS with multiple partners (at least two in a three-month period) was considered as 'higher-risk' sexual behaviour in this study as it may expose the person to the risk of contracting sexually transmitted infections (STIs), including HIV, thus affecting their health [46–48], and vice versa, CLS with only one partner or none in a three-month period was considered as 'lower-risk' behaviour. Furthermore, in our previous paper [29], we have reported that CLS2+ was strongly associated with a subsequent HIV incidence (it was a stronger association than for CLS), with a stronger association between time-updated CLS2+ and HIV incidence among men in the online cohort in AURAH2 (N = 622, aIRR 9.39, 95% CI 2.04–43.35, $p$ = 0.004). Guidelines from WHO published in 2018 also include unprotected sex with multiple partners in the past six months as criteria for PrEP to be indicated [52].

There are important limitations in this study. Because men in the AURAH2 study were recruited from three sexual health clinics in urban areas in southeast England between 2013

and 2018, they may not be typical of the overall GBMSM population in England and the United Kingdom and may not represent the trends in the current year. Trends in sexual behaviour and predictors of staying in or switching to higher-risk behaviours might also differ among GBMSM who are not engaged with sexual health clinics. Additionally, the sample comprised predominantly of men who were highly educated, employed, in a stable economic situation, of White ethnicity, and with access to the internet, which might not allow generalisability to all GBMSM living in England. The self-reported data may show recall bias and social desirability bias; nevertheless, the study obtained sensitive and personal data through an online follow-up questionnaire, which may have decreased desirability bias [53]. The online retention of participants who initially registered in the study was not optimal, and differences were observed among men who continued on the study and not. The individuals who dropped out were more vulnerable and socio-economically disadvantaged, suggesting differential bias. Therefore, the results in this study should be interpreted in this light of limitation. In the lasagna plot, entire-row sorting of results efficiently displayed the total number of outcomes. However, a disadvantage of sorting is that different outcomes are stacked on top of each other, which could lead to confusion. We included only two categories of sexual behaviour as including more categories in the plot would make interpretation increasingly challenging. The online phase of the study closed at the end of March 2018, and therefore some participants (who joined the study after March 2015) would not have been scheduled to receive a final questionnaire (if their previous questionnaire was less than four months preceding the end of follow-up). This resulted in the high proportions of skipping questionnaires in the last online questionnaire (Table 1 in Appendix 1). Finally, while Markov modelling can effectively depict changes in categorical risk behaviours over time, it is a computationally intensive technique and can be difficult to utilise when dealing with a large number of predictors, for example, due to concerns with statistical power and coefficient interpretation. Such limitations may hinder complicated model building. For introducing covariates, only data from participants with at least two consecutive follow-up data were utilised, thus limiting the capacity to relate analysis results to more descriptive transition probabilities predicted with more complete data.

In summary, our results demonstrated that men exhibited a variation in the trajectory of CLS2+ within an individual over time, with the tendency to remain consistent over four-month periods of time, and the majority of questionnaires reported lower-risk behaviour over time during the online follow-up phase of the study. This indicates that the majority of GBMSM are at low risk for HIV acquisition at any given point in time. Reducing HIV transmission may be possible if prevention strategies are directed towards GBMSM before an interval of elevated risk occurs.

## Supporting information

**S1 File. Longitudinal changes in condomless anal sex with multiple partners.**
(DOCX)

## Acknowledgments

We thank all study participants for their time and contributions. We gratefully acknowledge the three participating sites and the contributions and efforts of the following at each site:

The Mortimer Market Centre, London: Ana Milinkovic, Fabienne Styles, Rosana Laverick, Marzena Orzol, Emmi Suonpera

56 Dean Street Clinic, London: Ali Ogilvy

The Claude Nicol centre, Brighton: Celia Richardson, Elaney Youssef, Sarah Kirk, Marion Campbell, Lisa Barbour

## Author contributions

**Conceptualization:** Nadia Hanum, Valentina Cambiano, Janey Sewell, Alison J. Rodger, David Asboe, Gary Whitlock, Richard Gilson, Amanda Clarke, Ada R. Miltz, Simon Collins, Andrew N. Phillips, Fiona Lampe.

**Data curation:** Nadia Hanum, Valentina Cambiano, Janey Sewell, Ada R. Miltz.

**Formal analysis:** Nadia Hanum.

**Funding acquisition:** Alison J. Rodger.

**Methodology:** Nadia Hanum, Valentina Cambiano, Janey Sewell, Alison J. Rodger, David Asboe, Richard Gilson, Amanda Clarke, Ada R. Miltz, Andrew N. Phillips, Fiona Lampe.

**Supervision:** Andrew N. Phillips, Fiona Lampe.

**Writing – original draft:** Nadia Hanum.

**Writing – review & editing:** Nadia Hanum, Valentina Cambiano, Janey Sewell, Alison J. Rodger, David Asboe, Gary Whitlock, Richard Gilson, Amanda Clarke, Ada R. Miltz, Simon Collins, Andrew N. Phillips, Fiona Lampe.

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
