## [Decision Letter · Decision Letter 0]

21 Feb 2024

PONE-D-22-32487Transitions in sexual behaviour and the predictors among gay, bisexual, and other men who have sex with men in England: data from a prospective studyPLOS ONE

Dear Dr. Hanum,

Thank you for submitting your manuscript to PLOS ONE. After careful consideration, we feel that it has merit but does not fully meet PLOS ONE’s publication criteria as it currently stands. Therefore, we invite you to submit a revised version of the manuscript that addresses the points raised during the review process.

We look forward to receiving your revised manuscript.

Kind regards,

Anthony J. Santella, DrPH, MPH, MCHES

Academic Editor

PLOS ONE

Journal Requirements:

Additional Editor Comments:

As you can see from the below comments from the reviewers, there are a number of small, but important, areas across the different sections of the manuscript, that need either clarification and/or to be expanded. I feel these comments are not unreasonable and will result in a stronger, more publishable paper. 

Reviewers' comments:

Reviewer's Responses to Questions

**Comments to the Author**

1. Is the manuscript technically sound, and do the data support the conclusions?

Reviewer #1: Yes

Reviewer #2: Yes

2. Has the statistical analysis been performed appropriately and rigorously? 

Reviewer #1: Yes

Reviewer #2: Yes

3. Have the authors made all data underlying the findings in their manuscript fully available?

Reviewer #1: No

Reviewer #2: Yes

4. Is the manuscript presented in an intelligible fashion and written in standard English?

Reviewer #1: Yes

Reviewer #2: Yes

5. Review Comments to the Author

Reviewer #1: Many thanks for presenting me with this opportunity to review this paper. In the paper, the authors examine patterns of sexual behaviour and transitions between sexual behaviour levels among gay men in England. Within what is a somewhat unwieldly / lengthy paper there are a number of key findings of interest. Having read the paper a few times, I would suggest less is more in communicating these findings and making recommendations against them. Although of interest, the submission currently lacks a “so what”. Below I present a number of comments / suggestions to hopefully strengthen the submission. As a comment to the authors and journal, the version of the manuscript I received for first review had tracked changes presented.

Title

Consider a slight revision to the title to tie in / position “predictors” better – it currently floats somewhat independently.

Abstract

Rather than positioning the necessity for the research in relation to it being the first, it would be better to make clear the problem / need to be addressed – this is not made clear in the abstract (also a focus on being the first in the discussion which is unnecessary).

Does the journal require a structured abstract? If so, then sub-headers would be helpful.

Please make clear the annual cycle is in addition to the four monthly cycles (if this is indeed the case).

Be helpful to provide summary description of what consists as higher and lower risk and to be consistent in use of n/N and % - e.g. present 622 also as % of 1162.

Could you ensure the abstract results support directly the conclusion that “at any one point in time the majority of GBMSM are at low risk….”

The authors suggest their research “might” help identify people likely to increase their risk – consider rephrasing to highight how this analysis can inform progress.

Introduction

The first half of the introduction focuses on literature suggesting individual heterogeneity is important and that the “majority of data” (be good to consider another phrase or quantify) sits outside of the UK and that no previous study has been conducted. It would be helpful to utilise the introduction to provide supporting statements as to why such heterogeneity is important (what are the mechanisms, the challenges……?) and to focus less on where data are and more on the “so-what” of the issues being raised and the need for this exploration. The introduction currently feels slight.

It is unclear what the sentence about person-centred approaches adds – consider removing or repositioning.

The last paragraph says the same thing twice. Suggest focus less on what hasn’t been done and more on what is proposed here.

Methods

I assume at every third cycle of the four monthly questionnaire participants were expected to complete both the four monthly and annual questions – is that correct? Please make clear.

Under measures there is a discussion of the literature as well as a defence of the methods chosen – the review of the literature would be better placed in the introduction, and the defence better placed in the discussion.

Under measures it is stated that CLS2+ was the “main measure” of higher-risk – please make clear what the other measures were.

In the measure section we are informed that U=U was not widely “spread” and that PrEP use was “relatively” low – in the context of methods it is unclear why poorly defined and unreferenced statements such as these are being presented – please remove or amend so it is clear and supported.

I suggest the measures section be rewritten so reflects methods only and so that the approach taken is clear.

Under “Socio-demographic….” It would be helpful to add a little context as to how / why the variables listed are presented as a-priori predictors.

Results

As a general comment (and the comment that shifted my recommendation from minor to major revision), there is a lot to digest in the results. Given there is a good degree of overlap (for example, numerous ways of presenting changed states) in the results, and that methods are also included, please revise and reduce so that key findings are clear and potential actionable results (that are then deliberated on in the discussion) are focused on. Consolidate the different ways transition is tackled and presented and ensure there is a clear pathway through the results for the reader.

Small point, but no need for terms such as “Approximately” when presenting figures as specific as 12.2%

It would be transparent to present 622 and 411 as a % of 1162.

Helpful to compare those proceeding with those not-proceeding – hopefully in the discussion these numerous differences are discussed in depth in relation to limitations / interpretation.

In the 2nd paragraph of the results please rephrase so it is clear the comparison is 622 v 540 (currently presented only within brackets).

Again small point, in table 1 would be helpful to make clear 622 = completed baseline and at least one online follow-up questionnaire.

In trends of CLS2+ section it is suggested trends are shown among all 1162 men for 2013 to 2018 – however, I believe subsequent baseline data points were only available for 622 men, and of these only 411 engaged up to within final six months – please clarify? Also please make clear how there are 1665 baseline reports among the 1162 baselines (please make clear the two sets of figures are respective to baseline and follow-up).

I would suggest removing comment about decline up to end 2014 given paucity of data points and that you then state both measures rose from 2013. Given lack of data up to end 2014 consider censoring data up to then.

In relation to the sentence “….among 622 men, from the first online questionnaire to the last online questionnaire” should read “their” as not all 622 completed the first and last questionnaires in this study. Under this section on “Within person changes” please include denominators (e.g. in relation to excluding missing etc).

Please standardise definitions (e.g. either CLS2 or “CLS with two or more partners”), presentation of results (sometime only %, other times % and n/N, and also sometimes described qualitatively, e.g. “majority”, “appeared to be reasonably high stability”, “tended to decrease over time” & “relatively small”) and sub-headers (sometimes describe risk factors (e.g. within-person changes) and at other times tools (lasagna plot)). In a quantitative analysis such as this please qualify / quantify statements such as those listed above (i.e. “tended to decrease…” etc).

Throughout the results there is a fair amount of methodological rather than results-focused description of the figures – suggest ensuring the footnotes to each figures clearly explain what is being presented, and remove from results text, so to focus instead on the key findings.

A point of positioning – figure 3 presents the participants / time-point specific data that inform the more actionable summary results presented in figures 1 and 2 – consider first presenting figure 3 and then figures 1 and 2.

Discussion & conclusion

Remove repetition at start of discussions – I would suggest that there is no need to state your observed 2015 to 2018 trend was similar to your observed 2013 to 2018 trend.

Amend sentence “Among GBMSM with at least two consecutive data during follow-up…”

As with introduction, not clear why the issue of there potentially not having been a similar study in UK is repeated – please amend.

Although over 1500 words in the discussion focus on comparing the results of this analysis with the literature base, key findings are neither reflected on or have recommendations assigned against them – for example, having been a focus on the results, housing is only mentioned once in the discussion and then only to repeat what is presented in the results (no in-depth discussion or aligned recommendation for action).

The limitations needs to be expanded and I would suggest amending “there are some limitations” to something along the lines of “there are important limitations…” . Instead of focusing on the 60% of men enrolled in online follow-up who finished, please instead describe the limitation of 35% of men enrolled finishing. A reflection of the difference observed between the various enrolled populations is warranted. Given how much has changed since 2013 and 2018, please describe the potential limitation of these now somewhat aging data.

In addition to calling for future studies and suggesting nurses and clinicians “might” utilise this information, could more concrete actions / recommendations be put forward (whilst considering the important limitations of this study)?

Reviewer #2: I commend the authors for putting for a strong paper that aims to characterize "longitudinal patterns of sexual behaviour and determine transitions between sexual behaviour levels based on participants' reported condomless anal sex with two or more partners in a three-month period, among HIV-negative GBMSM participating in the AURAH2 study.

The study builds upon the existing AURAH questionnaire, limited due to its cross-sectional design, by allowing for longitudinal analyses. It is helpful to understand the strengths and limitations of AURAH2, which the authors did a fine job describing.

Minor corrections, if possible, include defining even the most obvious acronyms. The authors defined the United States (US) but not UK, for example. U=U should also include a definition.

If possible, the authors could elaborate on how missingness was addressed in greater detail and further provide a rationale for their decision. This should also be discussed in the limitations section.

6. PLOS authors have the option to publish the peer review history of their article (what does this mean? ). If published, this will include your full peer review and any attached files.

**Do you want your identity to be public for this peer review?** For information about this choice, including consent withdrawal, please see our Privacy Policy .

Reviewer #1: **Yes: ** Brian Rice

Reviewer #2: No

---

## [Author Response · Author response to Decision Letter 1]

10 May 2024

PONE-D-22-32487

Transitions in sexual behaviour among gay, bisexual, and other men who have sex with men in England: data from a prospective study

Response to the editorial team and reviewers

Dear Dr. Santella,

Thank you for the opportunity to submit a revised draft of our manuscript titled “Transitions in sexual behaviour among gay, bisexual, and other men who have sex with men in England: data from a prospective study” for consideration for publication at PLOS One.

We would like to thank the editorial team and reviewers for the careful and thorough reading of this manuscript and for the thoughtful comments and constructive suggestions, which helped to improve the quality of this manuscript. We have incorporated changes to reflect most of the comments provided by the reviewers. We have highlighted the changes within the manuscript using tracked changes, and the revised ‘clean’ manuscript is also included in this submission. All page and line numbers refer to the manuscript file with tracked changes. While there are major changes in our manuscript, the findings and conclusions are unchanged. Please see below, in blue, our point-by-point response to the reviewers’ and editors’ points and concerns.

As the corresponding author, I confirm that the manuscript has been read and approved for submission by all the named authors. Please address all correspondence concerning this manuscript to me at: nadia.hanum.17@ucl.ac.uk.

Thank you for your consideration of this manuscript. We look forward to hearing from you.

Sincerely,

Nadia Hanum

Comment from editor:

We've checked your submission and before we can proceed, we need you to address the following issues:

We note there may be a typo in the URL you provided for the National Research Ethics Service (NRES) in your Data Availability statement.

Please let us know if we may update your Data Availability statement to the following:

"Any personally identifiable data cannot be made publicly available as this study was conducted with approval from the National Research Ethics Service (NRES) committee, which only allows data from the studies to be released after the NRES provides written approval. This is in order to protect participants’ privacy. A de-identified dataset sufficient to reproduce the study findings can be made available upon written request, after approval from the NRES committee. To submit a request for these data, please contact nres.queries@nhs.net or go to www.nres.nhs/contacts/nres-committee-directory/."

If you approve, we will update this statement on your behalf.

Response: Thank you. We have renewed the URL provided and our data sharing statement. Apologies for the typos. Copied below is our changes to the data sharing statement:

Any personally identifiable data cannot be made publicly available as this study was conducted with approval from the National Research Ethics Service (NRES) committee, which only allows data from the studies to be released after the NRES provides written approval. This is in order to protect participants’ privacy. A de-identified dataset sufficient to reproduce the study findings can be made available upon written request, after approval from the NRES committee. To submit a request for these data, please contact hampstead.rec@hra.nhs.uk or queries@hra.nhs.uk or go to https://www.hra.nhs.uk/about-us/contact-us/.

Reviewer #1:

Many thanks for presenting me with this opportunity to review this paper. In the paper, the authors examine patterns of sexual behaviour and transitions between sexual behaviour levels among gay men in England. Within what is a somewhat unwieldly / lengthy paper there are a number of key findings of interest. Having read the paper a few times, I would suggest less is more in communicating these findings and making recommendations against them. Although of interest, the submission currently lacks a “so what”. Below I present a number of comments / suggestions to hopefully strengthen the submission.

Response: We thank reviewer #1 for the in-depth review and very detailed feedback. We have made changes according to the comments. Please see below.

- Title: consider a slight revision to the title to tie in / position “predictors” better – it currently floats somewhat independently.

Response: Thank you. We have removed ‘predictors’ from the title.

Abstract:

1) Rather than positioning the necessity for the research in relation to it being the first, it would be better to make clear the problem / need to be addressed – this is not made clear in the abstract (also a focus on being the first in the discussion which is unnecessary).

Response: Thank you. We have made changes in the abstract (lines 23 – 24).

2) Does the journal require a structured abstract? If so, then sub-headers would be helpful.

Response: Thank you. It does not say in the guidelines for authors that the abstract should have sub-headers.

3) Please make clear the annual cycle is in addition to the four monthly cycles (if this is indeed the case).

Response: Thank you. we have clarified this in the abstract (lines 30 – 32).

4) Be helpful to provide summary description of what consists as higher and lower risk and to be consistent in use of n/N and % - e.g. present 622 also as % of 1162.

Response: Thank you. Higher-risk and lower-risk have now been defined (between brackets) (lines 33 – 34) and we have presented 622 as % 1162 (line 36).

5) Could you ensure the abstract results support directly the conclusion that “at any one point in time the majority of GBMSM are at low risk….”

Response: Thank you. We have added some data in the abstract results that support our conclusion (lines 38 – 40).

6) The authors suggest their research “might” help identify people likely to increase their risk – consider rephrasing to highlight how this analysis can inform progress.

Response: Thank you. We have rephrased the conclusion (lines 50 – 51).

Introduction

1) The first half of the introduction focuses on literature suggesting individual heterogeneity is important and that the “majority of data” (be good to consider another phrase or quantify) sits outside of the UK and that no previous study has been conducted. It would be helpful to utilise the introduction to provide supporting statements as to why such heterogeneity is important (what are the mechanisms, the challenges……?) and to focus less on where data are and more on the “so-what” of the issues being raised and the need for this exploration. The introduction currently feels slight.

Response: Thank you. We agree with the reviewer. We have made changes in the introduction section so that it is now more focused on the importance of individual heterogeneity (lines 57 – 68). The unnecessary sentences have been removed (lines 68 – 72).

2) It is unclear what the sentence about person-centred approaches adds – consider removing or repositioning.

Response: Thank you. We have removed that part (lines 76 – 78).

3) The last paragraph says the same thing twice. Suggest focus less on what hasn’t been done and more on what is proposed here.

Response: Thank you. We have removed the sentence (lines 83 – 85).

Methods

1) I assume at every third cycle of the four monthly questionnaire participants were expected to complete both the four monthly and annual questions – is that correct? Please make clear.

Response: Thank you. Apologies for the confusion. Participants would only fill out the annual questionnaire at every third cycle. We have clarified this in the methods section (lines 114 – 120).

2) Under measures there is a discussion of the literature as well as a defence of the methods chosen – the review of the literature would be better placed in the introduction, and the defence better placed in the discussion.

Response: Thank you. We have moved the reasoning for choosing CLS2+ to the discussion section (lines 475 – 491).

3) Under measures it is stated that CLS2+ was the “main measure” of higher-risk – please make clear what the other measures were.

Response: Thank you. We have made clear in the sentence that CLS2+ is the only measure for higher-risk behaviour in this paper (lines 143 – 145).

4) In the measure section we are informed that U=U was not widely “spread” and that PrEP use was “relatively” low – in the context of methods it is unclear why poorly defined and unreferenced statements such as these are being presented – please remove or amend so it is clear and supported. I suggest the measures section be rewritten so reflects methods only and so that the approach taken is clear.

Response: Thank you. We have removed the sentence regarding low PrEP use and U=U (lines 155 – 160).

5) Under “Socio-demographic….” It would be helpful to add a little context as to how / why the variables listed are presented as a-priori predictors.

Response: Thank you. We considered all socio-demographic, health and lifestyle and other sexual behaviours variables collected in the study as possible predictors for the transitions in the CLS2+ level. We have made clear about this in the methods section (lines 163, 164, 167).

Results

As a general comment (and the comment that shifted my recommendation from minor to major revision), there is a lot to digest in the results. Given there is a good degree of overlap (for example, numerous ways of presenting changed states) in the results, and that methods are also included, please revise and reduce so that key findings are clear and potential actionable results (that are then deliberated on in the discussion) are focused on. Consolidate the different ways transition is tackled and presented and ensure there is a clear pathway through the results for the reader.

Response: Thank you. We appreciate and agree with reviewer #1 that some results overlap. We have reduced this section to only include important results and repositioned some figures and tables. Please see the detailed response below.

1) Small point, but no need for terms such as “Approximately” when presenting figures as specific as 12.2%

Response: Thank you. We have removed ‘approximately’ from the sentence (line 245).

2) It would be transparent to present 622 and 411 as a % of 1162.

Response: Thank you. We have corrected the % (line 249).

3) Helpful to compare those proceeding with those not-proceeding – hopefully in the discussion these numerous differences are discussed in depth in relation to limitations / interpretation.

Response: Thank you. We have added some discussion about this differential bias in the discussion section (lines 521 – 525).

4) In the 2nd paragraph of the results please rephrase so it is clear the comparison is 622 v 540 (currently presented only within brackets).

Response: Thank you. We have made clear about this comparison (lines 249 – 250).

5) Again small point, in table 1 would be helpful to make clear 622 = completed baseline and at least one online follow-up questionnaire.

Response: Thank you. this has been added to Table 1, third column.

6) In trends of CLS2+ section it is suggested trends are shown among all 1162 men for 2013 to 2018 – however, I believe subsequent baseline data points were only available for 622 men, and of these only 411 engaged up to within final six months – please clarify? Also please make clear how there are 1665 baseline reports among the 1162 baselines (please make clear the two sets of figures are respective to baseline and follow-up).

Response: Thank you. Yes, for the overall trends in CLS2+ data from all participants were used, even though not all participants had complete data until the end of follow-up. This has been stated in the footnote of Figure 1 and the number of questionnaires at every follow-up is also detailed there (now Figure 2). To account for repeated data within participants, we used the GEE model with logit link to assess trends among the 1162 men over the calendar year (statistical analysis sub-section lines 231 – 232). Regarding the 1665, that is the total baseline and follow-up questionnaires, not only baseline questionnaires, from the total 4,338 questionnaires provided by all participants. This has been stated in the footnote of Figure 2 and we have added this as well in the text (lines 366).

7) I would suggest removing comment about decline up to end 2014 given paucity of data points and that you then state both measures rose from 2013. Given lack of data up to end 2014 consider censoring data up to then.

Response: Thank you. We have removed the statement comment about the declining trend up to the end of 2014 (line 367).

8) In relation to the sentence “….among 622 men, from the first online questionnaire to the last online questionnaire” should read “their” as not all 622 completed the first and last questionnaires in this study. Under this section on “Within person changes” please include denominators (e.g. in relation to excluding missing etc).

Response: Thank you. This result (including Figure 2) has been removed from the manuscript as it depicts more or less the same results / overlaps with Figure 3 (now Figure 1).

9) Please standardise definitions (e.g. either CLS2 or “CLS with two or more partners”), presentation of results (sometime only %, other times % and n/N, and also sometimes described qualitatively, e.g. “majority”, “appeared to be reasonably high stability”, “tended to decrease over time” & “relatively small”) and sub-headers (sometimes describe risk factors (e.g. within-person changes) and at other times tools (lasagna plot)). In a quantitative analysis such as this please qualify / quantify statements such as those listed above (i.e. “tended to decrease…” etc).

Response: Thank you. We have made some changes regarding this matter (Please see throughout results section). We are trying to standardize the results, however there are several results which, apart from being in quantity, also have to be explained by qualifying the pattern, because this is mainly related to the explanation of images and trajectories.

10) Throughout the results there is a fair amount of methodological rather than results-focused description of the figures – suggest ensuring the footnotes to each figures clearly explain what is being presented, and remove from results text, so to focus instead on the key findings.

Response: Thank you. We have removed methodological contents from the results, for example texts in lines 266 – 268 have now been removed, and Figure 2 has been removed as well.

11) A point of positioning – figure 3 presents the participants / time-point specific data that inform the more actionable summary results presented in figures 1 and 2 – consider first presenting figure 3 and then figures 1 and 2.

Response: Thank you. We agree with the comment from reviewer #1. We have changed the order of the figures displayed; Figure 3 first, then Figure 1. Figure 2 is now not included in this manuscript as it depicts more or less the same results / overlaps with Figure 3.

Discussion & conclusion

1) Remove repetition at start of discussions – I would suggest that there is no need to state your observed 2015 to 2018 trend was similar to your observed 2013 to 2018 trend.

Response: Thank you. We have removed the second sentence about the similar trends between 2015-2018 and 2013-2018 (lines 439 – 445).

2) Amend sentence “Among GBMSM with at least two consecutive data during follow-up…”

Response: Thank you. This part has now been removed (Figure 2 and texts about within-person changes in the frequency in CLS2+ is no longer in the manuscript and we only show the lasagne plot as this more representative of the individual changes among men in our study).

3) As with introduction, not clear why the issue of there potentially not having been a similar study in UK is repeated – please amend.

Response: Thank you. We have also removed this sentence, as this is unnecessary.

4) Although over 1500 words in the discussion focus on comparing the results of this analysis with the literature base, key findings are neither reflected on or have recommendations assigned against them – for example, having been a focus on the results, housing is only mentioned once in the discussion and then only to repeat what is presented in the results (no in-depth discussion or aligned re

---

## [Decision Letter · Decision Letter 1]

10 Jun 2024

PONE-D-22-32487R1Transitions in sexual behaviour among gay, bisexual, and other men who have sex with men in England: data from a prospective studyPLOS ONE

Dear Dr. Hanum,

Thank you for submitting your manuscript to PLOS ONE. After careful consideration, we feel that it has merit but does not fully meet PLOS ONE’s publication criteria as it currently stands. Therefore, we invite you to submit a revised version of the manuscript that addresses the points raised during the review process. Although nearly ready to be accepted, the reviewer suggests some very minor comments which we kindly request to be addressed.

We look forward to receiving your revised manuscript.

Kind regards,

Avanti Dey, PhD

Staff Editor

PLOS ONE

Journal Requirements:

Additional Editor Comments (if provided):

Reviewers' comments:

Reviewer's Responses to Questions

**Comments to the Author**

1. If the authors have adequately addressed your comments raised in a previous round of review and you feel that this manuscript is now acceptable for publication, you may indicate that here to bypass the “Comments to the Author” section, enter your conflict of interest statement in the “Confidential to Editor” section, and submit your "Accept" recommendation.

Reviewer #1: All comments have been addressed

2. Is the manuscript technically sound, and do the data support the conclusions?

Reviewer #1: Yes

3. Has the statistical analysis been performed appropriately and rigorously? 

Reviewer #1: Yes

4. Have the authors made all data underlying the findings in their manuscript fully available?

Reviewer #1: Yes

5. Is the manuscript presented in an intelligible fashion and written in standard English?

Reviewer #1: Yes

6. Review Comments to the Author

Reviewer #1: I have recommended to accept this paper.

That said, I make here a couple of recommendations for further strengthening (not essential).

Another read through by the authors would be helpful just to ensure little errors are corrected, e.g. paragraph 1 of Introduction = “Identification of the behavioural windows associated with elevated HIV risk, i.e. the timely initiation and termination of such interventions is essential for successful prevention and to prevent medication overuse.”; 1st paragraph of results “(see table 1 [N]).”; 2nd paragraph of results “Of the 1,162 men enrolled, 622 (53.5%) men completed…”

Not critical, but the results, although much improved in structure, are still very long – if there is a way to further reduce / focus on key issues, then that would be helpful

Results (line 249; top of page 14) – could you qualify / quantify what is meant by “There appeared to be reasonably high stability among the group of men who reported lower-risk behaviour from one questionnaire to the next….”.

Results (lines 253 – 258) – “The high proportions of skipping questionnaires at the last online questionnaire was because not all participants had the opportunity to fill in the last online questionnaire. The online phase of the study closed at the end of March 2018, and therefore some participants (who joined the study after March 2015) would not have been scheduled to receive a final questionnaire (if their previous questionnaire was less than four months preceding the end of follow-up).” – could you please rephrase so clear what is being said and then move to your discussion as these are not results.

Results (line 291 – first sentence of new section entitled “Explanatory variables…” – sentence starts as “Similarly,……” – as new section, it is not clear what is being referred to – please rephrase.

Another read through of the discussion may be helpful to ensure small errors are removed (e.g. line 355 = “In the ACS, risk levels were classified into three (based on sexual behaviour score)” and to reduce the body of text (e.g. perhaps less repetition of the results) and break up long paragraphs and sentences.

7. PLOS authors have the option to publish the peer review history of their article (what does this mean? ). If published, this will include your full peer review and any attached files.

**Do you want your identity to be public for this peer review?** For information about this choice, including consent withdrawal, please see our Privacy Policy .

Reviewer #1: **Yes: ** Brian D Rice

---

## [Author Response · Author response to Decision Letter 2]

17 Jul 2024

Response: Thank you. We have reviewed all the references included and ensured that they are correct, and that no references that have been retracted are used. We have also adjusted the order of references according to those in the text.

Response: Thank you. We have done this step and the figures we included in this submission are in accordance with the adjustments made by PACE.

Comments from reviewer (Brian D Rice):

I have recommended to accept this paper. That said, I make here a couple of recommendations for further strengthening (not essential).

Response: We thank the reviewer for the positive feedback and the recommendation to accept our manuscript.

Another read through by the authors would be helpful just to ensure little errors are corrected, e.g. paragraph 1 of Introduction = “Identification of the behavioural windows associated with elevated HIV risk, i.e. the timely initiation and termination of such interventions is essential for successful prevention and to prevent medication overuse.”; 1st paragraph of results “(see table 1 [N]).”; 2nd paragraph of results “Of the 1,162 men enrolled, 622 (53.5%) men completed…”

Response: Thank you. We have made several changes related to what was suggested by the reviewer (Abstract section, line 31; Introduction section, lines 59 – 61; Results section, lines 215 – 216, 233 – 235, and 266).

Not critical, but the results, although much improved in structure, are still very long – if there is a way to further reduce / focus on key issues, then that would be helpful

Response: Thank you. We have followed the advice of the reviewer, to that end we have reduced the weight of our results to make it more concise. We have moved (previously) Table 2 and its explanatory text (Results section, lines 244 – 264) to Appendix 1. We have also made several changes to the text in Appendix 1 to make it clearer.

Results (line 249; top of page 14) – could you qualify / quantify what is meant by “There appeared to be reasonably high stability among the group of men who reported lower-risk behaviour from one questionnaire to the next….”.

Response: Thank you. We have made changes to this (Results section, lines 233 – 235)

Results (lines 253 – 258) – “The high proportions of skipping questionnaires at the last online questionnaire was because not all participants had the opportunity to fill in the last online questionnaire. The online phase of the study closed at the end of March 2018, and therefore some participants (who joined the study after March 2015) would not have been scheduled to receive a final questionnaire (if their previous questionnaire was less than four months preceding the end of follow-up).” – could you please rephrase so clear what is being said and then move to your discussion as these are not results.

Response: Thank you. We have rephrased and moved the sentences to the Discussion section (lines 472 – 476)

Results (line 291 – first sentence of new section entitled “Explanatory variables…” – sentence starts as “Similarly,……” – as new section, it is not clear what is being referred to – please rephrase.

Response: Thank you. We have removed ‘similarly’ from the first sentence of the section (line 294).

Another read through of the discussion may be helpful to ensure small errors are removed (e.g. line 355 = “In the ACS, risk levels were classified into three (based on sexual behaviour score)” and to reduce the body of text (e.g. perhaps less repetition of the results) and break up long paragraphs and sentences.

Response: Thank you. We agree with the reviewer. We have corrected some typos and errors, for example line 358 and reduced the text.

---

## [Editor Report · Decision Letter 2]

19 Jul 2024

Transitions in sexual behaviour among gay, bisexual, and other men who have sex with men in England: data from a prospective study

PONE-D-22-32487R2

Dear Dr. Hanum,

We’re pleased to inform you that your manuscript has been judged scientifically suitable for publication and will be formally accepted for publication once it meets all outstanding technical requirements. I would like to apologise for the delay in our communication.

Kind regards,

Daniel Demant, PhD, MPH, GradCertHEd, BAppSocSc

Academic Editor

PLOS ONE
---

## [Editor Report · Acceptance letter]

PONE-D-22-32487R2

PLOS ONE

Dear Dr. Lampe,

I'm pleased to inform you that your manuscript has been deemed suitable for publication in PLOS ONE. Congratulations! Your manuscript is now being handed over to our production team.

Kind regards,

on behalf of

Dr. Daniel Demant

Academic Editor

PLOS ONE